# Healthcare providers' perceptions on post abortion intrauterine contraception: A qualitative study in central Uganda

Herbert Kayiga[1]*, Emelie Looft-Trägårdh[2], Amanda Cleeve[2], Othman Kakaire[1], Nazarius Mbona Tumwesigye[3], Josaphat Byamugisha[1], Kristina Gemzell-Danielsson[2]

1 Department of Obstetrics and Gynecology, Makerere University College of Health Sciences, Kampala, Uganda, 2 Department of Women's and Children's Health, Karolinska Institutet, and WHO Collaborating Centre, Karolinska University Hospital, Stockholm, Sweden, 3 Makerere University School of Public Health, Makerere University College of Health Sciences, Kampala, Uganda

* hkayiga@gmail.com

**Data Availability Statement:** All the necessary data and materials underlying the findings described have been provided as part of this paper.

## Abstract

### Background

Despite access to post abortion intrauterine contraception, the uptake of Intrauterine devices (IUDs) in Uganda remains low. Whether the perceptions of healthcare providers towards IUDs have a role in the provision of post abortion IUDs remains unclear. We explored perceptions on post abortion IUD provision among healthcare providers in Uganda, focusing on barriers and facilitators in regards to provision and uptake.

### Methods

Between 1st August 2022 and 30th September 2022, forty-five in-depth interviews were conducted among healthcare providers of different cadres at sixteen public health facilities in central Uganda. We used the case study design to explore the healthcare providers' perceptions. The interviews were primarily to help us understand the perceptions of healthcare providers towards IUDs. All interviews were audio-recorded and transcribed verbatim. Themes were identified using the conventional inductive content analysis.

### Results

From the analysis, three themes emerged. Theme one covered health system related barriers in regards to IUD provision such as healthcare providers' and health facility challenges. The second theme focused on the challenges in post abortion contraceptive counselling focusing on IUDs. The third theme covered the motivating factors and participants' views on how to scale up IUD uptake and provision within post abortion care in Uganda. We found that lack of appropriate knowledge and skills on IUD provision, and heavy workloads, negatively impacted IUD provision. Inadequate facilities, IUD stock-outs, and minimal community sensitization also limited the utilization of IUDs. Furthermore, language barriers, community misconceptions around IUDs, long travel distances to the health facility, and partner refusal, contributed to the low uptake of post abortion IUDs. To address the identified barriers and

The authors agree to the free data sharing plan of the materials. In case any more data or materials are needed, they are readily accessible in the Department of Obstetrics and Gynecology library, Makerere University College of Health Sciences, Kampala, Uganda. The contact information at the Department of Obstetrics and Gynecology library, Makerere University College of Health Sciences, Kampala, Uganda is; Mrs. Maureen Ssekisolo The Department Secretary, mssekisolo2003@yahoo.com, mssekisolo200314@gmail.com, +256778442247.

**Funding:** This project was supported by funds from the Swedish Research Council, (Grant 2019-04256) in partnership with Makerere University and the MakRif Project. The content is solely the responsibility of the authors and does not necessarily represent the official views of Swedish Research Council, Makerere University or the MakRif Project. The funders provided support in the form of research expenses, but did not have any additional role in the study design, data collection and analysis, decision to publish, or preparation of the manuscript.

**Competing interests:** The authors have declared that no competing interests exist.

**Abbreviations:** CME, Continuing Medical education; DMPA, Injectable Depo-Medroxyprogesterone Acetate; IUC, Intrauterine Contraception; IUD, Intrauterine Device; FIGO, International Federation of Obstetricians and Gynecologists; LARCs, Long-Acting Reversible Contraceptives; LMICs, Low- and Middle-income countries; PAC, Post Abortion Care; mPAC, Medical Post Abortion Care; SOMREC, The School of Medicine Research and Ethics Committee; UNCST, Uganda National Council for Science and Technology; WHO, World Health Organization.

scale up post abortion IUD provision, participants recommended addressing health system barriers, regular in-service refresher trainings, mentoring and supervision. They emphasized the importance of addressing contraceptive misconceptions and men's opposition to IUDs through community sensitization.

## Conclusion

In this study we identified several barriers to post abortion IUD provision, highlighting an urgent need to address health system barriers including healthcare providers' skills and knowledge gaps, supply chain challenges, and to ensure that facilities are conducive to quality contraceptive counselling. Provision of on-job refresher trainings, mentoring and supervision, are key motivators that can be utilized in supporting healthcare providers towards post abortion IUD provision. To further increase uptake, efforts are needed to dispel contraceptive myths and misconceptions at the community level.

## Introduction

Unintended pregnancies continue to be a global public health issue especially in low- and middle-income countries (LMICs) [1, 2]. As a result of the high unmet need for contraception in LMICs between 20 to 58% [3], an estimated 45% of unintended pregnancies end up as induced abortions [4]. About 75.6% in Africa and 76.1% in East Africa of these induced abortions are unsafe, contributing to 5–13% of the preventable maternal mortality [5]. In sub-Saharan Africa, the high unmet need for contraception is driven by the healthcare providers' knowledge gaps in contraception, inaccessibility to modern contraceptive methods especially in rural areas, minimal resource allocation for family planning services and limited skills among healthcare providers [6].

Uganda has one of the highest maternal mortality ratios (of 189 per 100,000 live births) globally, and a young population with nearly 80% being 30 years or younger [7, 8]. Furthermore, Uganda has one of the highest fertility rates in the region at 5.2 children per woman [7]. With a high unmet need for contraception of 22% [7], 56% of the pregnancies in Uganda are unintended. About six percent of the maternal mortality in Uganda are due to abortion related complications [9], that follow the induced abortions in the face of restrictive abortion laws [10]. Uganda's contraceptive prevalence rate of 37% [7] is lower than that of her neighbors in the East African community; Kenya (58%) [11], and Rwanda (48.1%) [12] and Tanzania (38.4%) [13, 14]. Despite the availability of intrauterine devices in 90% of the health facilities, the national contraceptive prevalence rate of IUDs among married women in Uganda is two percent [7].

Prior studies have shown that fertility returns as early as five to ten days after first trimester abortions [15–17]. At the same time, over 50% of the women resume sexual intercourse within two weeks [18]. This highlights the importance of integrated contraceptive services in comprehensive post abortion care and early initiation of contraception for those who wish to prevent or delay another pregnancy. Existing research from the sub-Saharan context shows that many women leave post abortion care without having received contraceptive counselling or method [19].

There's uncertainty on whether the low uptake of post abortion IUDs in Uganda, is driven by the perceptions of the healthcare providers, limitations in the skill base, or the medicolegal

environment [20]. According to Benson et al in ten countries in Asia and Sub-Saharan Africa [21], trained healthcare providers in intrauterine contraception were more likely to recommend post abortion contraception as compared to the healthcare providers who were not trained. Whether it is the healthcare providers' knowledge or skill gaps in the provision of post abortion intrauterine contraception to explain the low uptake is not well documented.

Prior studies in Uganda on post abortion contraception have highlighted the benefits of offering long-acting reversible contraception like IUDs in reducing the six percent of the maternal deaths resulting from abortion related complications [9, 20, 22]. Despite the benefits that come with the IUDs, Ugandan women remain skeptical of IUDs as a result of misconceptions and biases from the IUD associated complications. These studies recommended revamping healthcare providers' skills and knowledge base to improve the uptake of the post abortion intrauterine contraception [20, 22]. Information about healthcare providers perspectives on post abortion contraceptive counselling and provision may shed light on barriers and solutions to ultimately improve care and increase uptake of effective contraceptives. It is against this background that this study set out to explore the views and perceptions of healthcare providers towards the provision of post abortion IUDs.

## Materials and methods

### Study design

This qualitative study was part of a broader project on post abortion IUD insertion in central Uganda. We used the relativist ontological approach as there are different perceptions from the different healthcare providers based on their background training, context-bound personal experiences and expertise in the provision of family planning services at the different health facilities. We used the case study design and the conventional inductive content analysis approach to answer the study question [23].

### Study setting

This study was conducted at sixteen public health facilities in central Uganda. The selected facilities offer free medical and contraception services five-days a week. These facilities are run by healthcare providers of different cadres, with a senior midwife overseeing the administrative duties. These health facilities were purposively selected because they were the biggest family planning service providers in central Uganda. The facilities also employed healthcare providers of different cadres offering services in family planning and emergency gynecological care such as medical and surgical evacuation of incomplete abortions (S1 Annex).

### Participant recruitment and sampling

With the guidance of the heads of the family planning units and the medical superintendents at the different health facilities, healthcare providers offering gynecological services based on their availability and convenience, were invited to participate in the study. We aimed to recruit healthcare providers irrespective of their cadres actively involved in post abortion care services at the sixteen selected health facilities during the study period who consented to participate in the study. Potential participants were provided with the study information in-person or over phone. They then suggested a time of their convenience with preference for afternoons after most of the days' duties. We purposively selected 2–3 participants to ensure variability in terms of cadre, age, clinical experience and sex, per facility. Participants included heads of the family planning units, obstetricians/gynecologists, nurse-midwives, and medical doctors, in the family planning clinics. The selected healthcare providers were given two contacts, of the

principal investigator and the research team, to consult at any point when they needed more information about the study.

**Data collection procedure.** *Staff training.* The research team consisted of three groups with experience in qualitative enquiry. Group one was assigned to collect data. This selected team was familiar with the local hospital settings. Prior to data collection, they received training in the study procedures including recruitment and informed consent procedures while observing the research ethics in accordance to the Declaration of Helsinki [24]. Group two was in charge of data analysis. This team had to ensure transcription accuracy and data analysis. Group three was made of two independent researchers whose task was to ensure rigor in the study according to the Lincoln–Guba criteria [25].

*Study materials.* We used a semi-structured interview guide consisting of open-ended questions covering the socio-demographic information, perceptions on PAC provision and post abortion contraceptive counselling, barriers and facilitators with regards to provision of post abortion IUDs. The interview guide was piloted prior to data collection during interviews with four conveniently sampled healthcare providers. Next, we made modifications to three questions in the interview guide to ensure that the tool captured the study objectives and appropriateness of the language.

*Data collection.* In-depth interviews were conducted between 1$^{st}$ August 2022 and 30$^{th}$ September 2022. All interviews were conducted in English, the official language in Uganda. The interviews were conducted by one interviewer and note taker in order to capture non-verbal expressions and field notes. The interviews were conducted in quiet rooms free from interference at the different health facilities. The interviews were conducted in the afternoons after much of the day's work was completed to minimize interference from patient care. In view of the fact that most of the healthcare providers recognized the first author (HK) a gynecologist, he exempted himself from the interviews to allow free participants' expression during the interviews. He was however actively involved in the briefing of the team before and debriefing thereafter to streamline the subsequent interviews. We took notes during the interviews for triangulation. All interviews were recorded and transcribed verbatim. After ascertaining that no new data was emerging from the interviews, we ascertained richness or saturation, and stopped the data collection thereafter [26, 27]. The interviews lasted between 45 to 90 minutes. Whenever clarity was needed, more specific questions were raised by the interviewer so that all the required information was collected.

*Rigor.* To ensure rigor in the data collection, we used Guba and Lincoln criteria [25], that included, data credibility, confirmability, transferability and dependability. Triangulation was by interviewing different cadres of healthcare providers from different health facilities and later on comparing their perspectives. Triangulation was ensured by group three (HK, RM) that was devoted to continued reading through of the transcripts to ensure ongoing comparison of the key information generated from one health facility to another. We ensured prolonged data collection. Prolonged engagement with the data was observed thereafter to make appropriate interpretation. A journal where constant reflections of our experiences, potential biases and assumptions was kept. In so doing we kept the findings in perspective with the study assumptions from the outset of the study. We also observed the individuals' posture and body language and made notes that we compared with the other data from the interviews. Dependability was observed by the stringent coding procedure and inter-coder corroboration. Key concepts and themes arising from the data were organized into codes through an iterative process of reading the interview transcripts for recurring patterns. From the insights, initial codes were developed that were later on transformed, and labelled. A codebook from the codes was generated thereafter. Data confirmability was observed by ensuring that participants' statements were captured with barely any modifications made. Data transferability was ensured by

the research team so that a rich, condensed description of the study process was documented to enable replicability in a similar context elsewhere [28]. To strengthen trustworthiness, we built rapport with the participants and also assured them of confidentiality. We also had variation in the cadres interviewed to assess whether our findings were in agreement or not. We ensured that the data collection team was different from those actively involved in the implementation of the study. We used interview probes in the interviews and ensured that the settings of the interview rooms were appropriate to minimize these potential biases from the participants.

## Data analysis

Using both manifest and latent content analysis [29], the data collected was analyzed inductively by HK, and RM [30]. The steps taken were as follows; [23] the interviews were transcribed, into sentences, phrases and paragraphs. Meaning units were generated from the content of the participants' voices. The meaning units were then condensed. We then converted the condensed meaning units into codes. We then compared the codes based on their similarities or differences and a codebook was created. Similar codes were organized into subcategories and these subcategories were later put into categories. The categories were later organized into themes. In cases of disagreements, the research team had to discuss until an agreement was reached. This was ensured by Group three (RM, HK).

## Framework analysis

We used the Socio-ecological model (SEM) [31] as the theoretical framework developed by Bronfenbrenner in 1979, to understand the perceptions of the healthcare providers towards the provision of post abortion intrauterine contraception. The SEM framework was used to identify barriers, facilitators and recommendations towards provision of post abortion intrauterine contraception by the healthcare providers at individual, inter-personal, institutional and societal levels [32]. At individual level, we anticipated that healthcare providers' misconceptions [33], inadequate knowledge and skill sets [34] to be potential barriers towards post abortion intrauterine contraception. Continued professional development and support supervision [21], were identified as facilitators at individual level towards recommendation of post abortion intrauterine contraception. At inter-personal level, peer influence to recommend other contraceptive methods over IUDs [33], was anticipated to limit the provision of post abortion IUDs. At institutional level, stock-outs of the different family planning choices, heavy patient turnovers, competing responsibilities for the healthcare providers [32, 35], were anticipated as potential barriers. Among the anticipated facilitators at institutional level included; team work, continued reskilling of healthcare providers, availability of family planning supplies and materials, and the integration of family planning services with abortion care [32]. At the societal level, myths and misconceptions on IUDs, transportation limitations, knowledge gaps [6], restrictive abortion laws [10] and policy gaps [36] on the provision of post abortion intrauterine contraception, were anticipated as potential barriers. We anticipated this SEM framework would guide in identification of gaps in the current practice that could be used to formulate evidence-based multilevel interventions to improve access and uptake of post abortion intrauterine contraception.

## Ethical considerations

Ethical approvals were obtained from The Makerere University School of Medicine Research and Ethics Committee, (Mak-SOMREC-2021-131), and Uganda National Council for Science and Technology (HS2111ES). Administrative clearances were obtained from the sixteen health

facilities. Verbal and written informed consents were obtained from all study participants prior to data collection. Participants were reassured that participating in the study was voluntary and that they could opt out of the study without compromising the relationship with the research team and service delivery. Participants were compensated for their time. Confidentiality and participants' rights were observed throughout the study. All transcripts were pseudonymized and data (audiotapes, records, transcripts and notes) were kept in a secure location accessible only to the study personnel.

## Results

Between 1st August 2022 and 30th September 2022, we interviewed forty-five healthcare providers of different cadres including obstetricians/gynecologists, nurse-midwives, nurses and administrators. The average age of the participants was 38(±6.5) years. Majority of the participants had Bachelors' degrees and higher degrees, however, a few had diplomas in nursing or midwifery. The great majority of healthcare providers had more than five years' experience offering contraception services (Table 1).

From the content analysis, three themes, emerged. Theme one covers participants' perceptions of barriers to post abortion IUD uptake and provision at the health system level. The

**Table 1. Socio-demographic characteristics of 45 healthcare providers from 16 health facilities in central Uganda.**

| Characteristic | Number of participants (n) | Percentage (%) |
|---|---|---|
| **Gender** | | |
| Female | 42 | 93.3 |
| **Age (years)** | | |
| 25–29 | 10 | 22.2 |
| 30–34 | 6 | 13.3 |
| 35–39 | 2 | 4.4 |
| 40–44 | 14 | 46.7 |
| 45–49 | 8 | 17.8 |
| 50+ | 5 | 11.1 |
| **Education** | | |
| Certificate | 7 | 15.5 |
| Diploma | 10 | 22.2 |
| Bachelors | 21 | 46.7 |
| Master's Degree | 7 | 15.5 |
| **Marital Status** | | |
| Single | 15 | 33.3 |
| Married | 30 | 66.7 |
| **Religion** | | |
| Catholic | 13 | 28.9 |
| Protestant | 10 | 22.2 |
| Pentecostal | 9 | 20 |
| Adventist | 3 | 6.7 |
| Muslim | 7 | 15.6 |
| Atheist | 3 | 6.7 |
| **Contraception experience (years)** | | |
| <1 | 4 | 8.9 |
| 1–5 | 10 | 22.2 |
| 6–10 | 13 | 28.9 |
| >10 | 18 | 40.0 |

**Table 2. Overview of themes and categories of the healthcare providers' perceptions on the uptake of post abortion intrauterine contraception in central Uganda.**

| Theme | Categories | Condensed meaning units |
|---|---|---|
| 1. Perceived Health system barriers to post abortion IUD provision | Perceptions on Healthcare providers' barriers | 1. Counselling for IUD use more demanding than for short-acting methods. |
| | Perceptions on Health facility barriers | 2. Inadequate supply of materials and other supplies. 3. There was limited space for provision of IUD services. 4. Lack of knowledge and skills. |
| 2. Challenges in counselling women on post abortion IUDs | Perceptions on patient level barriers: | 1. IUDs can cause infertility, cancers and death 2. Financial constraints 3. Partners oppose IUD use 5. IUDs involve invading women's privacy to have them inserted. 4. Some of the vocabulary used for hormonal IUDs are non-existent in the local languages. |
| 3. Participants' views on factors that facilitate post abortion IUD provision and scaling up | Facilitators for provision of post abortion IUDs. | 3. Having Continuous professional development in post abortion IUD use. 2. Mentorship and supervision are pertinent to motivating healthcare providers to offer IUD services |
| | Recommendations to scale up IUD uptake and provision. | 3. Training on how to counsel women on post abortion IUD use. 4. Use of social media platforms to sensitize communities on post abortion IUDs. |

second theme looks at the challenges in post abortion contraceptive counselling and specifically on IUDs. The third theme focuses on participants' views on factors that would facilitate post abortion IUD provision and scale up post abortion IUD uptake in Uganda (Table 2).

Participants were of the view that post abortion contraception faces barriers mainly from within the health system, themselves as the healthcare providers and the patients. These included language barriers, knowledge and skill gaps in IUD insertions, heavy patient loads, inadequate facilities, and lack of materials and IUD equipment, and resulted in healthcare providers often opting for short-acting contraception instead of LARCs. Participants noted that misconceptions around IUDs, socio-cultural pressure to have children shortly after a miscarriage or abortion, spousal disapproval to IUD use, and patients' limited access to IUD services impede the uptake of post abortion IUDs in central Uganda. Upon recognizing these barriers, the participants identified solutions that would improve IUD provision and uptake, such as in-service refresher trainings, mentoring and supervision, maintaining IUD materials and equipment, as well as community sensitization on IUD use to dispel myths and misconceptions.

## Theme 1: Perceptions of healthcare system barriers in regards to post abortion IUD provision

The participants reported that provision of post abortion IUDs hinges on health system barriers that need to be addressed, including barriers at the level of the providers and at the facility level. Participants highlighted some of the challenges that they encounter as they provide post abortion IUDs, such as unavailability of the necessary IUD equipment and supplies, limitations in the infrastructure, overwhelming patient load, and limited human resource in the provision of post abortion IUDs.

**Barriers at healthcare provider level.** Among the healthcare providers' perceived barriers included, lack of appropriate knowledge and skills in the IUD insertion, patient overload, inadequate time for post abortion family planning counselling, lack of knowledge regarding initiation of post abortion contraception.

*Lack of appropriate knowledge and skills in IUD insertion.* Majority of the participants reported lack of skills and knowledge on the insertion of the post abortion IUDs. This was

because of the fact that insertion was done in particular sections, especially the Family Planning clinics that run parallel to emergency clinics where abortions happened. Those who had not worked in the emergency gynecology wards or clinics, were not well versed with the procedure of inserting the IUDs. Besides, some if not all the participants had not seen hormonal IUDs, therefore, getting challenged. Such challenges compelled them to opt for short-acting methods like injectables (DMPA).

> *We lacked knowledge and skills. There are those things that you could not thoroughly give details and then somebody takes it right. There are some myths that I could not prove wrong before. I know we are taught throughout the medical training but if you are not practicing it, one would prefer giving a method that she is competent with. We need to do refresher training courses to update people on the most current practices that are safe for the clients.* **Female, 26 years 02 GBE**

*Inadequate time for post abortion contraceptive counselling.* Most of the participants acknowledged that implementing this method of family planning required more time. They explained how counselling must be comprehensive in order for the women to make an informed decision. However, due to the nature of the work at the health facilities, participants were caught up with other routines like delivering mothers which were more of an emergency than the provision of the intrauterine contraception.

> *Because even now we are experiencing it. Someone says, 'When you are to insert an IUD, the counselling you need to give takes more time compared to may be if you are to offer pills or Depo-injection. And again, the information that is needed to be provided to the client is a bit longer compared to the one for Depo-injection. And since it is not a commonly used method by the clients, so they really need time to make them accept the method.* **Female, 42 years 01 KYG**

> *When you get a client, you should counsel her comprehensively so that she understands the benefits. The woman has to also understand the return of her fertility. They also need to understand the likely risks and outcomes they may face when they conceive very early.* **Female, 38 years 03 KGD**

**Barriers at the health facility level.** *Stock-outs, lack of equipment and inadequate facilities.* Stock-outs were reported as a major barrier to post abortion contraceptive provision. This is because of irregularities in the supply chain system where the healthcare providers have nothing to do. This hindered the implementation because some of the women would come and ask for the method yet it was not available.

> *Sometimes the IUDs are not available. That was before when those NGOs came and trained us, we had them but we were not trained. And then there was a time they were giving the hormonal type and yet they had serious side effects, so they stopped giving them out until they changed to another hormonal.* **Female, 41 years 01 NGH**

Participants reported that some of the general wards and units lacked some family planning commodities that were exclusively the mandate of the Family Planning units. Women who presented during weekends or on public holidays, didn't receive any contraceptive counselling because the family planning units were closed. It was also reported that other wards like the gynecology or labour wards lacked the materials needed for the intrauterine contraception.

*Particularly in my ward, we didn't have the methods themselves. We could talk about them but then send them to either (Maternal-child health unit) MCH or any other unit that has family planning. The methods were not available at our unit as (post abortion care) PAC, so after managing women with the pill or the evacuation, we would just tell them to go to MCH. So, it was still hindered because we couldn't offer the services since the methods were not available on the ward.* **Female, 26 years 02 GBE**

Most health facilities also lacked the necessary equipment to use in the implementation of post abortion IUDs. Some health facilities only had one IUD set and there were always delays in sterilizing the equipment before they could be used for the next available clients. This created long waiting times and some women would leave without a method. Some participants also reported limitations in regard to where to offer family planning services, because the same allocated space was used for counselling, triaging and treatment for gynecological emergencies in some health facilities.

*We don't have enough IUD sets. Sometimes, you have used the only available sterile set but you have four women waiting for the IUDs. The other challenge we face is one of space for the IUD procedures. We use the same procedure beds for delivery and the IUD insertion. This becomes so challenging when we have a mother in labour.* **Female, 41 years 01 NGH**

*Inadequate human resource.* Participants reported inadequate number of staff that could offer contraceptive services and heavy patient loads, preventing provision of quality contraceptive counselling.

*Yeah, by the way, inadequate staffing is a problem. We are few staff and you have to cover different shifts. If somebody can be here delivering a mother, you have a line of mothers and you are very few on duty. So that is also a big challenge. And sometimes it's challenging to work alone, you are tired and there is a woman requesting for family planning. You might not serve that woman well because you are tired. Sometimes we go past our time for work yet we stay far. It is really a challenge.* **Female, 31 years 01 KWH**

## Theme 2: Challenges in counselling women on post abortion IUDs

Participants relayed several challenges in counselling women on IUDs. For instance, participants described several myths and misconceptions about IUDs among women seeking PAC that were difficult to dispel. This was complicated by language barriers with some words around hormonal IUDs nonexistent in local languages. Other barriers to uptake included women's desire to heal before the IUD was inserted, partner refusal, inaccessibility of the IUD services, and social pressure to conceive. All these highlighted factors mounted as perceived challenges to have women accept and initiate post abortion IUDs.

**Myths and misconceptions.**   One of the major challenges in contraceptive counselling described by participants in regards were myths and misconceptions about the IUD among patients. They reported that women had deep-rooted myths that the IUDs cause cancers, reduce sexual pleasure, make the men uncomfortable during sexual intercourse and even disappear in their bodies and move up to the heart causing death, as illustrated in the following excerpt.

*I told you about the myths. Women think that when you insert the IUD, it will go as far as where the uterus is and it will get to the heart causing death! Some say that when you insert it,*

*the husband can feel that she is using family planning. Though you counsel them that you will not feel the threads, they feel that the husband will feel them. Then another issue is, after counselling the women, about the myths on IUDs, some think if it is inserted, they will never get pregnant again. Then others say that many women will get cervical cancer or will be perforated, so they fear the IUDs.* **Female, 42 years 03 KYG**

Participants described how some women wrongly associated vaginal discharge during IUD use, with infections, that led to IUD removals.

*Of course, also the infections, they tend to... You know sometimes you can have a change in the discharge, not necessarily that it is an infection. But every time they have increased discharge they tend to come, "oh, I have been having a bad discharge," and they...so they associate it with infections and then the partners. Sometimes you even go ahead to check and you find they don't have an infection. It is just an increase in the discharge.* **Female, 30 years 02 KST**

**Motivating women to return for IUD insertion.** Participants revealed several challenges in motivating women to return for the IUD insertion. They described how many women wanted time to heal before initiating a contraceptive method and felt uncomfortable with vaginal examinations while they were still bleeding. Another challenge reported by the participants was that many women have long travel distances to the healthcare facility and therefore fail to come back. This led some participants to recommend immediate insertion of the IUD for those who wanted one. Not all women come back for follow-up and when efforts to call them for review, their numbers don't always go through:

*Sometimes they are okay, they don't feel anything. In our country, we don't have that habit of coming to health facilities when you are not sick. So, when the women feel okay, they don't come back. Others may not have understood what you told them or they forget, or others don't have phones. We usually get their phone contacts, and when we check in the registers about their appointment, we call them. Quite often the phone contacts are wrong and never get through. This makes it very hard to connect to these women.* **Male, 34 years 01 KGD**

Some of the participants reported that women are not interested in the IUD because the procedure of inserting it is viewed as invasive and compromised their personal integrity. This made it challenging to promote the IUD as a suitable post abortion method.

*There are these women who are less interested in the IUDs I would say. Ever since I learnt how to insert the IUD, I have gotten very few women who take up that method of family planning. You know, ladies looking in their private parts, they fear. They would prefer it but having to put up their legs and having the IUDs inserted is an uphill task. The process of inserting IUDs makes them usually refuse the method.* **Female, 43 Years 01 KST**

Participants also reported that some women want to conceive as fast as possible following treatment of an incomplete abortion. This was described as due to pervasive pronatalist norms in Uganda and relationship dynamics in polygamous families.

*Another thing, some of the women are eager to get pregnant again because of the pressure they have from their husbands or the pressure due to the polygamous families. They are*

*competing to give birth with their co-wives. When you tell her to use a method, they will show you that they don't want to use any method. She wants to have a baby to save her marriage because the husband has put her under pressure.* **Female, 34 years 01 KWL**

**Partner opposition to IUDs.** Some participants depicted husbands as an important barrier to post abortion IUD uptake, describing how men in general were against their wives using IUDs. In cases where the husband accompanied the woman in the hospital, the husband could decline IUD provision on behalf of the woman or make her return for removal.

*May be about the partners, because some partners do not want the methods. Here, most of the clients come by themselves. They don't come with their spouses. So, when you give a method to a woman, she goes and again the husband brings her back, 'remove this thing, I don't want it.' So, it is a bit challenging.* **Female, 34 years 01 KGD**

**Language barriers.** Participants highlighted language barriers as a challenge in contraceptive counselling. The healthcare providers at times lacked sufficient local terminologies to explain to the potential users of the IUDs.

*Sometimes it's the language barrier because you have to make the person understand. You know English! you explain the methods very well but in the local language, it is difficult to get some words like 'hormones!' Some people will fail to understand what a hormone is.* **Female, 41 years 01 NGH**

## Theme 3: Facilitators and recommendations for provision of post abortion IUDs

Overall, participants were passionate about providing good quality contraceptive counselling. They emphasized the importance of women having access to effective methods, leading to healthier women and babies. Participants discussed how different factors in the work environment enabled provision of good quality care and identified motivators that could help address challenges in post abortion contraceptive counselling. They recommended on-job refresher training, ongoing supervision and mentorships, and ensuring that IUD equipment and supplies were available. The motivators and recommendations, as suggested by the participants, are narrated below.

**An enabling work environment.** In-service training was emphasized as an important motivator that improved providers skills and knowledge. Practicing IUD insertion on a regular basis gave healthcare providers more experience and confidence in post abortion IUD insertion. Supervision from more experienced providers kept them motivated and encouraged them to keep up the practice. Furthermore, they noted that continuous professional development was essential for improving the provision and uptake of post abortion IUDs.

*That continuous monitoring, and support, which wasn't there in the past, keeps us going. What motivates me even further, is us being supervised. When people come and supervise us and they see what we are doing, and correct us when we are going wrong. As it is a daily procedure, you can reach somewhere and when you are stuck, yet your fellow nurse is not here, you call and then be guided.* **Female,38 years 02 KYG**

Several participants reported how positive collegial relations and teamwork made work more enjoyable and enabled more women to have the IUD inserted.

*May be another motivator is the teamwork we are having. Because if I leave a client here, I can call (Name), and say, "(mentions name again), I have left a mother, help insert for me.' I have consented, and screened her but I have not finished the other gap of insertion. So that one also gives you motivation. Because you feel you are not alone.* **Female, 34 Years 01 KGD**

To further increase women's access to post abortion contraceptive methods, participants stressed the need to address healthcare provider shortages, contraceptive stockouts and other health system barriers that currently impede contraceptive services. They underscored the need for more healthcare providers with adequate training to provide post abortion IUDs, to ensure that this method was always available to those who wanted it, but also to reduce workloads and limit waiting times for patients.

*More training is needed for the healthcare providers. Not all of us are skilled to insert the IUDs. We need more skills and knowledge about the insertion of IUDs. To keep on updating us, to keep on training us, is the most important task, so that we are more skilled and we become experts.* **Female, 34 years 01 KWL**

**Community sensitization on post abortion IUDs.** A recommendation that was frequently mentioned by participants, was community sensitization post abortion IUDs. The participants suggested using channels like radio and television announcements and programs concerning the family planning services. They also recommended the use of social media channels to reach the people like TikTok, and Facebook among others. This would not only make people aware but they would also give evidence-based information and dispel myths and misconceptions about IUDs.

*Then another thing we need is comprehensive sensitization at all levels, at the facility, community, government levels, and district, like radio talk shows, information in the newspapers, and community dialogue can be held in the community on IUDs. Then may be encouraging in understanding health for self, women to understand their health themselves which helps them to make healthy choices.* **Female, 30 years 02 KST**

## Discussion

This qualitative study sought to explore the perceptions of healthcare providers on provision of post abortion IUDs, at sixteen public facilities in central Uganda. Our study identified health systems and healthcare providers' barriers in utilization of post abortion intrauterine contraception. The study also explored challenges reaching out to provide post abortion contraceptive counselling to women managed for first trimester incomplete abortions in central Uganda. Upon recognizing these barriers, the participants identified solutions that would improve IUD provision and uptake, such as in-service refresher trainings, mentoring and supervision, maintaining chain supply of IUD materials and equipment, as well as community sensitization on IUD use to dispel myths and misconceptions.

We found how health system barriers constitute a major barrier to good quality post abortion contraceptive counselling and provision. This is in line with what previous studies have acknowledged as similar barriers resulting in provision of short-acting methods even for

women with long-term needs [37–40]. Our findings highlight several solutions to the identified barriers that require long-term investments in health system strengthening. Uganda has always depended on external donors to facilitate family planning services with the United States, one of the largest funders [41]. Though the President of Uganda pledged five million USD annually in 2020 towards the commitment of family planning to reduce the unmet need for contraception to ten percent and increase the contraceptive prevalence to 50%, the financing of the family planning services hasn't been sufficient to achieve this target [41]. Furthermore, with the reduction in the international funding in Uganda towards family planning services, the strain on the already overwhelmed health system is gross, leading to an overburden on the limited human resource and infrastructure [41]. This could be the explanation for the barriers as identified in the current study.

In several countries in sub-Saharan Africa, IUD user rates remain below one percent [42]. The uptake of IUDs among Ugandan women has also been reported at 3.8% by Twesigye [43]. In the United States of America, even when the IUDs are readily accessible, the current percentage of users between ages 15–49 years was at 14% [44] despite the reported effectiveness and satisfaction rates [45]. The utilization of IUDs in São Paulo, Brazil, was reported to be 1.7 percent [46]. Prior studies globally show that healthcare providers could have an influence on the lower uptake despite the proven effectiveness of IUDs [34, 47–49]. Our findings expose a lack of knowledge and skills among healthcare providers, which made them promote post abortion short-acting contraceptives over LARCs. Similar findings have been reported in prior studies globally [34, 47–49]. In Egypt [50], 82.5 percent of the healthcare providers reported to have negative attitudes towards IUDs.

Our findings point to misconceptions about IUDs, among women obtaining PAC, as a barrier to uptake. Other studies have also identified misconceptions about the IUD among healthcare providers like IUDs causing pelvic infection, which discouraged them to recommend IUDs to the potential users [51]. Only ten percent of the healthcare providers in the study conducted in China recommended the use of IUDs among the adolescents and unmarried nulliparous women [51]. In Uganda, the knowledge and skillset gaps among healthcare providers have been reported as the reason for the IUD contraceptive prevalence of less than four percent [20, 43].

Our findings show how important in-service training and supervision for maintaining healthcare providers' skills and confidence relates to IUD insertion. According to a study by Madden in Saint Louis, United States of America [34], healthcare providers who had freshly graduated or saw more contraceptive patients were more likely to offer IUDs as compared to those who graduated before 1989 or saw fewer contraceptive patients. This emphasizes the necessity of on-job or in-service healthcare providers' refresher trainings in post abortion IUD provision.

Our findings suggest that myths and misconceptions among women obtaining PAC remain a major barrier for uptake and are perceived as a major challenge to be addressed by healthcare providers. Twesigye also reported similar thought patterns among Ugandan women towards IUDs in 2016 [43]. In Egypt, nulliparous women in addition to fearing the subfertility associated with the IUDs, also expressed that the fact that healthcare providers had to insert the IUDs in their bodies, interfered with their privacy. The associated bleeding and pain at insertion, made 96.2 percent of the participants to shun the IUDs [50].

Our findings reinforce the notion of men as barriers to contraceptive use in this context. Male disapproval has been documented as a limiting factor to contraceptive uptake in sub-Saharan Africa by Blackstone [35]. Previous research from the United States, show that men were less knowledgeable about IUDs as compared to the other methods. Men who were

Christians and had never had any prior sexual health visits were even less knowledgeable about IUDs. Such men were more likely to discourage their spouses on using IUDs [52].

The strength of this study lies in the diversity of cadres that were involved in understanding the experiences, and perceptions of healthcare providers towards post abortion IUDs. We interviewed forty-five participants from sixteen public health facilities in rural, peri-urban and urban areas in central Uganda. We conducted in-depth interviews as opposed to focus group discussions that allowed individual healthcare providers to freely express their views as opposed to fear of what their peers would think of their views.

Since the study was conducted only in the public health facilities where all care was free, our findings may not be transferable to the private health system setting. Furthermore, as this study was nested in a larger project on post abortion IUDs, healthcare providers in included healthcare facilities had received training and supervision regarding post abortion IUD provision. The perspectives of participants in our study regarding the specific type of challenges they face may therefore differ from that of healthcare providers in other public health facilities in Uganda. The barriers, and motivators for insertion of IUDs are different from those of the public facilities. Without a national scheme that offer all medical care free of charge in all health facilities in Uganda, whether cost of care for post abortion IUDs had an impact on the perceptions of healthcare providers towards post abortion IUDs couldn't be explored in the study. Though the healthcare providers reported that patient factors were pertinent to provision of post abortion IUDs, women's perceptions or experiences in this regard, were not explored in this study for triangulation.

## Conclusion

By exploring healthcare providers' views and perceptions, we identified several barriers to post abortion IUD provision in Uganda. Our findings highlight an urgent need to address health system barriers to improve quality of post abortion contraceptive counselling and enhance provision of post abortion IUDs. Measures to address healthcare providers' gaps in knowledge skills relating to post abortion IUD provision should be prioritized. Community misconceptions, and partner opposition were highlighted as important barriers to IUD uptake that must also be addressed. Provision of on-job refresher trainings, availability of equipment and materials, mentoring and supervision, would improve quality of post abortion contraception, including IUD provision and uptake.

## Supporting information

**S1 Annex. Background characteristics of study sites.**
(DOCX)

**S1 File. Approved protocol.**
(PDF)

## Acknowledgments

We appreciate the research team and study participants for making this study a dream come true. We are indebted to Mr. Richard Muhumuza (RM), Diane Achanda Genevive (DAG), and Diana Nankabirwa (DN) for the study coordination, critiquing the interview guide, and spearheading the data collection and analysis. A special vote of thanks goes to the Administrators of the health facilities that participated in the study. I'm so humbled by the support and guidance accorded to us by the Doctoral committee (Associate Prof Annettee Nakimuli, Dr

Joseph Rujjumba, Dr Musa Sekikubo) and my PhD supervisors at Makerere University, Kampala Uganda.

## Author Contributions

**Conceptualization:** Herbert Kayiga, Emelie Looft-Trägårdh.

**Formal analysis:** Herbert Kayiga, Othman Kakaire.

**Funding acquisition:** Herbert Kayiga, Kristina Gemzell-Danielsson.

**Methodology:** Herbert Kayiga, Emelie Looft-Trägårdh, Amanda Cleeve, Othman Kakaire, Nazarius Mbona Tumwesigye, Josaphat Byamugisha, Kristina Gemzell-Danielsson.

**Supervision:** Amanda Cleeve, Othman Kakaire, Nazarius Mbona Tumwesigye, Josaphat Byamugisha, Kristina Gemzell-Danielsson.

**Writing – original draft:** Herbert Kayiga.

**Writing – review & editing:** Herbert Kayiga, Emelie Looft-Trägårdh, Amanda Cleeve, Othman Kakaire, Nazarius Mbona Tumwesigye, Josaphat Byamugisha, Kristina Gemzell-Danielsson.

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
