## [Decision Letter · Decision Letter 0]

10 Jun 2024

PONE-D-24-05509Healthcare Providers’ Perceptions on Post Abortion Intrauterine Contraception: A qualitative study in central UgandaPLOS ONE

Dear Dr. Kayiga, Thank you for submitting your manuscript to PLOS ONE. After careful consideration, we feel that it has merit but does not fully meet PLOS ONE’s publication criteria as it currently stands. Therefore, we invite you to submit a revised version of the manuscript that addresses the points raised during the review process.

We look forward to receiving your revised manuscript.

Kind regards,

Yitagesu Habtu Aweke, Ph.D

Academic Editor

PLOS ONE

Journal Requirements:

"I received part of the funding from Kalorinska Institutet in partnership with Makerere University for the data collection and analysis but not in towards the publication and dissemination. I would be grateful if I can be availed a full publication waiver towards my publication.

I have in the past reviewed peer articles for free for this highly reputable journal and would be grateful if my request for a publication waiver is given your kind consideration."

Reviewers' comments:

Reviewer's Responses to Questions

**Comments to the Author**

1. Is the manuscript technically sound, and do the data support the conclusions?

Reviewer #1: Yes

Reviewer #2: Partly

Reviewer #3: Yes

2. Has the statistical analysis been performed appropriately and rigorously? 

Reviewer #1: Yes

Reviewer #2: No

Reviewer #3: No

3. Have the authors made all data underlying the findings in their manuscript fully available?

Reviewer #1: Yes

Reviewer #2: No

Reviewer #3: Yes

4. Is the manuscript presented in an intelligible fashion and written in standard English?

Reviewer #1: Yes

Reviewer #2: Yes

Reviewer #3: Yes

5. Review Comments to the Author

Reviewer #1: Summary of the Research

Despite access to post abortion intrauterine contraception, the uptake of Intrauterine devices (IUDs) in Uganda is below four percent. Whether the perceptions of healthcare providers towards IUDs have a role in the provision of post abortion IUDs remains unclear. The aim of study to explore the perceptions of healthcare providers towards post abortion IUDs after medical management of first trimester incomplete abortions. Methods: Between 1st August 2022 and 30th September 2022, forty-five in-depth interviews were conducted among healthcare providers of different cadres in sixteen public health facilities in central Uganda. The interviews were primarily to help us understand the perceptions of healthcare providers towards IUDs. All interviews were audio-recorded and transcribed verbatim. Themes were identified using inductive content analysis. From the content analysis of the interviews, three themes emerged. Theme one covered health facility related barriers such as healthcare providers’ and health system challenges towards provision of post abortion IUDs. The second theme was around the challenges in reaching and counselling women to use the post abortion IUDs. The third theme covered the motivating factors and participants’ views on how to scale up IUD uptake and provision within postabortion care in Uganda. Even with the availability of the necessary materials and equipment required for post abortion IUDs, and improved logistics such as timing of insertion to overcome some of the barriers faced by patients, service delivery hinges on the healthcare providers’ perceptions. Measures to address healthcare providers’ knowledge gaps and IUD skill sets should be enhanced. Healthcare providers’ misconceptions need to be addressed with provision of evidence-based information to enable correct information to patients. There is a need to address health system barriers like patient overload, stock-outs of equipment and materials for IUD insertion, and privacy. Provision of on-job refresher trainings, community sensitization on IUDs, financial incentives, and support supervision are key motivators that can be utilized in supporting healthcare providers towards post abortion IUD provision.

Areas for improvement

Title:

Good title.

Abstract:

Clear and good abstract.

Key words: limits the words to the most important, 5-6 words is enough.

Order of the authors:

Mention all authors of this review please.

Make sure that all necessary information for all authors are available in this document.

Introduction Section:

In one sentence explain if maternal death is associated with not using birth spacing methods.

The authors should revise introduction section for grammar issues and language to improve readability.

Join paragraphs that discussing the same concepts.

The authors should indicate what they wanted the readers to understand.

Overall

Good introduction but the authors need to arrange introduction section.

Material and Methods:

Study design,

Clear.

Setting,

Clear.

Participants recruiting,

Clear.

Inclusion and exclusion criteria,

Clear.

Quality Assessment,

The authors should explain the how the maintained validity of data.

Data collection and analysis,

Clear, however topic guild should be explained in detail with the questions.

Ethical consideration,

Clear.

Results:

The authors should revise the language and grammar to improve readability.

Discussion:

The authors should make clear for the readers about the themes that they were discussing.

The authors should discuss each themes from different angle and in detail.

The authors should revise the language and grammar to improve readability.

References:

The authors should revise all references according to the guidelines provided.

Overall: Good manuscript, all the best.

Reviewer #2: This study describes the perceptions of healthcare providers in Central Uganda on post-abortion intrauterine contraception. However, the inclusion of only 45 interviewees does not adequately represent the broader situation in Central Uganda. The study employs inductive content analysis, but this method also has some shortcomings, particularly in terms of replicability, bias impact, and theoretical depth. The subjectivity of the researchers may affect which data are coded and how they are coded, thereby influencing the objectivity and reliability of the research results. Researchers may tend to select data that supports their anticipated hypotheses while ignoring or undervaluing data that does not fit their expectations. This could lead to the research results being biased or incomplete.

Reviewer #3: The manuscript is entitled: "Healthcare Providers’ Perceptions on Post-Abortion Intrauterine Contraception: A Qualitative Study in Central Uganda'".

The authors explored the healthcare providers' perceived barriers and facilitators to post-abortion IUDs after medical management of first-trimester incomplete abortions in Central Uganda.

General Comment: The research topic is relevant to the field and the authors thoroughly explored the research topic and demonstrated a deep understanding of the subject matter. However, during a review of the manuscript, the following areas were identified as a major concern

1. The authors clearly described the gap in low uptake of IUDs in Uganda, however, they failed to define the evidence gap. Specifically, evidence gap on the health providers' perceptions of post-abortion IUD. What others presented on the topic needs to be well presented. So, What were the gaps in the literature?

2. The title of the manuscript describes healthcare providers' Perceptions, however, what is presented in the result, discussion and even in the conclusion part is entirely about the barriers and facilitators to post-abortion IUD uptake?? It lacks coherence. The ideas should flow logically from one section to the next, with clear transitions. Each section should build upon the previous one, guiding the reader through the manuscript.

3. The reviewer can not say the methodology is sound and replicable with confidence

3.1. Despite having a clear study design, the authors exhaustively discussed the philosophical perspective that aligns with this study. "A qualitative interview study" is not specific. Which kind of qualitative study design was employed to answer the research question???

3.2. The authors assigned two independent researchers whose task was to ensure rigor in the study according to the Lincoln–Guba criteria. That's excellent. However, how was trustworthiness (rigor) ensured in this study? Specifically, what techniques were used to establish credibility, confirmability, dependability, and transferability?

Additionally, the following minor revisions/comments were identified:

1. Abstract

1.1. Methods: what was analyzed and the type of analysis conducted?

1.2. Results: Content analysis appeared in the result part. wouldn't it be repeating the methods in the result section?

2. Results

2.1. the title of Table 1: why Baseline characteristics?

2.2. Table 2: is not self-explanatory and is very long

3. Discussion: Unnecessary details were observed in this section, therefore, the authors need to be more focused while discussing only the main findings. Moreover, very long sentences need to modified again.

6. PLOS authors have the option to publish the peer review history of their article (what does this mean?). If published, this will include your full peer review and any attached files.

Reviewer #1: **Yes: **Zalikha Khamis Al-Marzouqi

Reviewer #2: No

Reviewer #3: **Yes: **Andamlak Gizaw Alamdo

---

## [Author Response · Author response to Decision Letter 0]

17 Jul 2024

COLLEGE OF HEALTH SCIENCES

SCHOOL OF MEDICINE

DEPARTMENT OF OBSTETRICS AND GYNAECOLOGY

15th July 2024

RESPONSE TO REVIEWS COMMENTS:

Object: PONE-D-24-05509: “HEALTHCARE PROVIDERS’ PERCEPTIONS ON POST ABORTION INTRAUTERINE CONTRACEPTION: A QUALITATIVE STUDY IN CENTRAL UGANDA” 

With great pleasure, I’m thankful for your comments towards our manuscript. In response to the reviewers’ comments sent to us on 11th June 2024, we have revised the manuscript accordingly.

Comment Response to Comment Page No. and Line 

Journal Requirements:

1. Please ensure that your manuscript meets PLOS ONE's style requirements, including those for file naming. The PLOSONE style templates can be found at 

“ I received part of the funding from Kalorinska Institutet in partnership with Makerere University for the datacollection and analysis but not in towards the publication and dissemination. I would be grateful if I can be availed a full publication waiver towards my publication.

I have in the past reviewed peer articles for free for this highly reputable journal and would be grateful if my request for a publication waiver is given your kind consideration."

Reviewers' comments:

Reviewer's Responses to Questions

Comments to the Author

1. Is the manuscript technically sound, and do the data support the conclusions?

Reviewer #1: Yes

Reviewer #2: Partly

Reviewer #3: Yes

2. Has the statistical analysis been performed appropriately and rigorously? 

Reviewer #1: Yes

Reviewer #2: No

Reviewer #3: No

3. Have the authors made all data underlying the findings in their manuscript fully available?

Reviewer #1: Yes

Reviewer #2: No

Reviewer #3: Yes

4. Is the manuscript presented in an intelligible fashion and written in standard English?

Reviewer #1: Yes

Reviewer #2: Yes

Reviewer #3: Yes

5. Review Comments to the Author

Please use the space provided to explain your answers to the questions above. You may also include additional comments for the author, including concerns about dual publication, research ethics, or publication ethics. (Please upload your reviewas an attachment if it exceeds 20,000 characters)

Reviewer #1: Summary of the Research

Despite access to post abortion intrauterine contraception, the uptake of Intrauterine devices (IUDs) in Uganda is below four percent. Whether the perceptions of healthcare providers towards IUDs have a role in the provision of post abortion IUDs remains unclear. The aim of study to explore the perceptions of healthcare providers towards post abortion IUDs after medical management of first trimester incomplete abortions. Methods: Between 1st August 2022 and 30th September 2022, forty-five in-depth interviews were conducted among healthcare providers of different cadres in sixteen public health facilities in central Uganda. The interviews were primarily to help us understand the perceptions of healthcare providers towards IUDs. All interviews were audio-recorded and transcribed verbatim. Themes were identified using inductive content analysis. From the content analysis of the interviews, three themes emerged. Theme one covered health facility related barriers such as healthcare providers’ and health system challenges towards provision of post abortion IUDs. The second theme was around the challenges in reaching and counselling women to use the post abortion IUDs. The third theme covered the motivating factors and participants’ views on how to scale up IUD uptake and provision within postabortion care in Uganda. Even with the availability of the necessary materials and equipment required for post abortion IUDs, and improved logistics such as timing of insertion to overcome some of the barriers faced by patients, service delivery hinges on the healthcare providers’ perceptions. Measures to address healthcare providers’ knowledge gaps and IUD skill sets should be enhanced. Healthcare providers’ misconceptions need to be addressed with provision of evidence-based information to enable correct information to patients. There is a need to address health system barriers like patient overload, stock-outs of equipment and materials for IUD insertion, and privacy. Provision of on-job refresher trainings, community sensitization on IUDs, financial incentives, and support supervision are key motivators that can be utilized in supporting healthcare providers towards post abortion IUD provision.

Areas for improvement

Title:

Good title.

Abstract:

Clear and good abstract.

Key words: limits the words to the most important, 5-6 words is enough.

Order of the authors:

Mention all authors of ffthis review please.

Make sure that all necessary information for all authors are available in this document.

Introduction Section:

In one sentence explain if maternal death is associated with not using birth spacing methods.

The authors should revise introduction section for grammar issues and language to improve readability.

Join paragraphs that discussing the same concepts.

The authors should indicate what they wanted the readers to understand.

Overall

Good introduction but the authors need to arrange introduction section.

Material and Methods:

Study design,

Clear.

Setting,

Clear.

Participants recruiting,

Clear.

Inclusion and exclusion criteria,

Clear.

Quality Assessment,

The authors should explain the how the maintained validity of data.

Data collection and analysis,

Clear, however topic guild should be explained in detail with the questions.

Ethical consideration,

Clear.

Results:

The authors should revise the language and grammar to improve readability.

Discussion:

The authors should make clear for the readers about the themes that they were discussing.

The authors should discuss each themes from different angle and in detail.

The authors should revise the language and grammar to improve readability.

References:

The authors should revise all references according to the guidelines provided.

Overall: Good manuscript, all the best.

Reviewer #2: This study describes the perceptions of healthcare providers in Central Uganda on post-abortion intrauterine contraception. However, the inclusion of only 45 interviewees does not adequately represent the broader situation in Central Uganda. 

The study employs inductive content analysis, but this method also has some shortcomings, particularly in terms of replicability, bias impact, and theoretical depth. The subjectivity of the researchers may affect which data are coded and how they are coded, thereby influencing the objectivity and reliability of the research results. Researchers may tend to select data that supports their anticipated hypotheses while ignoring or undervaluing data that does not fit their expectations. This could lead to the research results being biased or incomplete.

Reviewer #3: The manuscript is entitled: "Healthcare Providers’ Perceptions on Post-Abortion Intrauterine Contraception: A Qualitative Study in Central Uganda'".

The authors explored the healthcare providers' perceived barriers and facilitators to post-abortion IUDs after medical management of first-trimester incomplete abortions in Central Uganda.

General Comment: The research topic is relevant to the field and the authors thoroughly explored the research topic and demonstrated a deep understanding of the subject matter. However, during a review of the manuscript, the following areas were identified as a major concern

1. The authors clearly described the gap in low uptake of IUDs in Uganda, however, they failed to define the evidence gap. Specifically, evidence gap on the health providers' perceptions of post-abortion IUD. What others presented on the topic needs to be well presented. So, What were the gaps in the literature?

2. The title of the manuscript describes healthcare providers' Perceptions, however, what is presented in the result, discussion and even in the conclusion part is entirely about the barriers and facilitators to post-abortion IUD uptake?? It lacks coherence. The ideas should flow logically from one section to the next, with clear transitions. Each section should build upon the previous one, guiding the reader through the manuscript.

3. The reviewer can not say the methodology is sound and replicable with confidence

3.1. Despite having a clear study design, the authors exhaustively discussed the philosophical perspective that aligns with this study. "A qualitative interview study" is not specific. Which kind of qualitative study design was employed to answer the research question???

3.2. The authors assigned two independent researchers whose task was to ensure rigor in the study according to the Lincoln–Guba criteria. That's excellent. However, how was trustworthiness (rigor) ensured in this study? Specifically, what techniques were used to establish credibility, confirmability, dependability, and transferability?

Additionally, the following minor revisions/comments were identified:

1. Abstract

1.1. Methods: what was analyzed and the type of analysis conducted?

1.2. Results: Content analysis appeared in the result part. wouldn't it be repeating the methods in the result section?

2. Results

2.1. the title of Table 1: why Baseline characteristics?

2.2. Table 2: is not self-explanatory and is very long

3. Discussion: Unnecessary details were observed in this section, therefore, the authors need to be more focused while discussing only the main findings. Moreover, very long sentences need to modified again.

6. PLOS authors have the option to publish the peer review history of their article (what does this mean?). If published, this will include your full peer review and any attached files.

Do you want your identity to be public for this peer review? For information about this choice, including consent withdrawal, please see our Privacy Policy.

Reviewer #1: Yes: Zalikha Khamis Al-Marzouqi

Reviewer #2: No

Reviewer #3: Yes: Andamlak Gizaw Alamdo

The manuscript has been revised in accordance to the PLOSONE style templates as recommended.

All the necessary data and materials underlying the findings described have been provided as part of this manuscript. The authors agree to the free data sharing plan of the materials. In case any more data or materials are needed, they are readily accessible in the Department of Obstetrics and Gynecology library, Makerere University College of Health Sciences, Kampala, Uganda.

The manuscript has been revised in accordance to the PLOSONE style templates. We have added a caption for the supporting information as recommended.

Thanks to the reviewers for considering the manuscript to be technically sound.

We appreciate the reviewers for the comments in this regard. We have availed the revised manuscript, the approved protocol, ethical approval forms from the School of Medicine, Makerere University Ethic Review, and the Uganda National Committee of Science and Technology, informed consent and interview guide for the in-depth interviews. We have also attached the descriptive characteristics of the study sites without exposing the identity of the participants. The authors agree to the data sharing plan.

Thanks to the reviewers for considering the manuscript worthy in this regard.

The key words used in the revised manuscript are now five. They are as follows; First-trimester abortion, Intrauterine-contraception, Perceptions, Healthcare 

---

## [Decision Letter · Decision Letter 1]

15 Aug 2024

Healthcare Providers’ Perceptions on Post Abortion Intrauterine Contraception: A qualitative study in central Uganda

PONE-D-24-05509R1

Dear Dr. Herbert Kayiga,

We’re pleased to inform you that your manuscript has been judged scientifically suitable for publication and will be formally accepted for publication once it meets all outstanding technical requirements.

Kind regards,

Yitagesu Habtu Aweke, Ph.D

Academic Editor

PLOS ONE

**Comments to the Author**

1. If the authors have adequately addressed your comments raised in a previous round of review and you feel that this manuscript is now acceptable for publication, you may indicate that here to bypass the “Comments to the Author” section, enter your conflict of interest statement in the “Confidential to Editor” section, and submit your "Accept" recommendation.

Reviewer #3: All comments have been addressed

2. Is the manuscript technically sound, and do the data support the conclusions?

Reviewer #3: Yes

3. Has the statistical analysis been performed appropriately and rigorously? 

Reviewer #3: N/A

4. Have the authors made all data underlying the findings in their manuscript fully available?

Reviewer #3: Yes

5. Is the manuscript presented in an intelligible fashion and written in standard English?

Reviewer #3: Yes

6. Review Comments to the Author

Reviewer #3: (No Response)

7. PLOS authors have the option to publish the peer review history of their article (what does this mean?). If published, this will include your full peer review and any attached files.

Reviewer #3: **Yes: **Andamlak Gizaw Alamdo

---

## [Editor Report · Acceptance letter]

26 Aug 2024

PONE-D-24-05509R1 

PLOS ONE

Dear Dr. Kayiga, 

I'm pleased to inform you that your manuscript has been deemed suitable for publication in PLOS ONE. Congratulations! Your manuscript is now being handed over to our production team.

Kind regards, 

on behalf of

PhD Candidate Yitagesu Habtu Aweke 

Academic Editor

PLOS ONE